

# The impact of side and top arm techniques during the backstroke breakout phase on 15-meter swimming performance

Zhenyu Jin[1,2], Yuhang Zhou[2,3], Dapeng Wang[1,2] and Yuhong Wen[2,3]

[1] College of Education, Beijing Sport University, Haidian, Beijing, China
[2] Key Laboratory of Sport Training of General Administration of Sport of China, Haidian, Beijing, China
[3] School of Recreational Sports and Tourism, Beijing Sport University, Haidian, Beijing, China

Corresponding author
Yuhong Wen, wyhswim@126.com

## ABSTRACT

**Background:** Research on the swimming starts and turns in professional swimming has become increasingly refined. The breakout phase is a crucial transition from point between underwater and above-water movements. The side arm technique is commonly used during the backstroke breakout phase. However, some swimmers have also achieved good performance using the top arm technique. The impact of the two techniques during the backstroke breakout phase is yet to be explored.

**Purpose:** To compare the velocity and key angles differences between the side arm and top arm techniques and analyze their effect on the first 15-m performance after push-off.

**Methods:** The subjects were 16 high-level swimmers: eight males (20.4 ± 1.6 yr) and eight females (20.9 ± 1.7 yr). Of these, 14 were backstrokers or had individual medley as their primary event. The best performances in the history of the sport's level World Aquatics Points are 682.1 ± 59.0 and 729.3 ± 41.5 for males and females, respectively. A within-subject design was used to test both the side arm and top arm techniques. Four underwater cameras were used to capture two-dimensional data from two perspectives, focusing on the breakout phase. A two-way ANOVA used to compare segment velocity, angles, and 15-m performance between the two genders and breakout techniques. Pearson's correlation analysis was used to explore the relationship between segment velocity and 15-m performance, and hierarchical regression was employed to investigate the impact of breakout velocity on the 15-m performance.

**Results:** Both male and female swimmers exhibited a significantly faster top arm velocities of (1.72 ± 0.20 and 1.47 ± 0.16 m/s, respectively) compared to the side arm velocities (1.51 ± 0.23 and 1.29 ± 0.19 m/s, respectively) during the breakout phase. Additionally, significant differences were found between the two genders ($F = 11.189$, $p = 0.002$, $\eta^2 = 0.286$) and breakout techniques ($F = 8.014$, $p = 0.008$, $\eta^2 = 0.223$), with no interaction effect ($F = 0.037$, $p = 0.849$, $\eta^2 = 0.001$). Furthermore, both male ($R = -0.447$, $p = 0.109$) and female ($R = -0.555$, $p = 0.017$) swimmers showed a moderate positive correlation between breakout velocity and 15-m performance, and the regression model indicated a significant impact on the 15-m performance.

**Conclusion:** The top arm technique during backstroke may offer a velocity advantage over the traditional side arm technique during the breakout phase, influencing 15-m performance. However, considering the short duration of the

breakout phase, this advantage and its impact may need to be considered in conjunction with the smoothness of the transition to the subsequent phases.

# INTRODUCTION

Start and turn performance is critical for a good swimming performance, the first 15-m distance is one of the most important bases for delineating the start and turn phases is the 15 m after the start or push-off (turn) (*Morais et al., 2019*; *Gonjo & Olstad, 2021*; *Morais et al., 2022a*; *Barbosa et al., 2021*; *Born et al., 2024*). Based on the differences in movement patterns, these phases can be further subdivided into glide and underwater propulsion (dolphin kick) stages. Previous studies have demonstrated the significant contribution of these sub-phases to the start or turn phase, as well as the overall swimming performance (*Marinho et al., 2020*; *Tor, Pease & Ball, 2015*; *Fischer & Kibele, 2016*; *West et al., 2022*; *Trinidad, 2024*).

The breakout phase is integral to the transition from underwater to above-water movements in both the start and turn phases of swimming, the impact of the breakout phase on the overall race time of high-level swimmers has gained increasing research attention (*Stosic et al., 2020*). The significance of this phase is reflected in both the breakout velocity and rhythm of subsequent swimming velocity and movements (*Veiga & Roig, 2017*; *Shen et al., 2015*; *Stosic et al., 2023*).

The definition of the breakout phase differs across studies, and its characteristics vary among different strokes (*Stosic et al., 2020*; *Trinidad et al., 2020*). Certain studies have harmonized the breakout phase to include the underwater and/or clean swimming phases (*Takeda et al., 2009*; *Trinidad et al., 2022*; *Stosic et al., 2023*). Based on these studies, we employed the following definition of the backstroke breakout phase: the first arm movement during the start and turn phases when swimmers transition from underwater to above-water state, excluding arm recovery, starting from hand separation to hand push completion.

In traditional backstroke techniques, the stroke movements during the breakout phase are generally consistent with those in the subsequent clean swimming phase or can be equivalent to the first arm movement in the underwater phase. However, some parameters, like stroke frequency (SF) and stroke length (SL), may differ significantly from those of the underwater dolphin kick and/or clean swimming phases (*Stosic et al., 2020*; *Shen et al., 2015*). Certain leading backstroke swimmers adopt a technique involving above-body arm movements during the start and turn phases, which increases the side tilt angle of the body, thereby achieving desirable results, however there is a relative lack of research references. This technique has not been widely adopted by backstroke swimmers, and its efficiency needs to be empirically studied.

The study according to the position of the arms relative to the body for the two backstroke breakout techniques as top arm and side arm. In the side arm technique, the

stroke arises from the side of the body during the breakout phase; it can be interpreted as the first backstroke stroke action to an extent. Meanwhile, in the top arm technique, the stroke arises from above the body during the breakout phase. In this study, we compared the velocities and angles of the two techniques and analyzed the impact of their breakout phase velocities on the 15-m performance of the swimmers. Based on the results presented for different swim strokes and swimmer and the practical use of the two techniques, the following hypotheses are proposed: (1) A significant difference exists in the speed and technique of the side and top arm strokes in the breakout phase; (2) the speed difference of the two strokes during the breakout phase is not decisive for start and turn performance; and (3) the speed difference of the two strokes is not decisive for the start and turn performance.

## MATERIALS AND METHODS

### Participants

This study included 16 high-level swimmers, including eight males (20.4 ± 1.6 yr) and eight females (20.9 ± 1.7 yr), with >8 yr of training experience. Of these, 14 were backstroke/individual medley swimmers (400/200 m) whose primary/secondary event was a short-to medium-distance event (200/100/50 m). According to the World Aquatics Points, the best performance in a major event was 682.1 ± 59.0 for males and 729.3 ± 41.5 for females (further information provided in Table 1). As not all participants specialized in backstroke, we recorded only the 50-m backstroke performances. In case a participant did not compete in the 50-m backstroke, their performance during training or test (conducted near the study date) was considered for analysis.

The participants were selected as follows: (1) Volunteers; (2) those who had participated in competitions with a minimum of 600 World Aquatics Points for the individual's best-ever performance in the main event (50-m pool, backstroke/individual medley events; however, this was not mandatory as the individuals were judged and screened in conjunction with the test performance, regardless of the event distance); (3) those who swim >20,000 m per week or participate in at least five training sessions totaling up to 10 h (testing recent minimum requirements); (4) individuals who are free from any injuries/ illnesses that can affect test performance; (5) participants who did not undergo any intense swimming/strength training 2 d prior to the test; and (6) participants who were informed about the test movements 1 month prior to the test, who had practiced the movements during training or were capable of performing the top arm breakout technique during the test. Signed informed consent was obtained from the participants. The research procedure complied with the Declaration of Helsinki and was approved by the Sports Science Experiments of the Beijing Sport University, Beijing, China (approval number: BSU2023223H).

### Process

The tests were conducted in a 50-m standard swimming pool with a water temperature of 27.5 °C. According to the individual competition routines, all the participants were given 20–45 min for warm-up, including land and water exercises. The total swimming distance

**Table 1 Descriptive statistics of the basic information of the subjects.** M ± SD represents Mean ± one standard deviation.

| Characteristics | M ± SD | |
| --- | --- | --- |
| | Male (*n* = 8) | Female (*n* = 8) |
| Age (years) | 20.43 ± 1.57 | 20.92 ± 1.74 |
| Height (m) | 1.83 ± 0.05 | 1.74 ± 0.04 |
| Body mass (kg) | 79.54 ± 5.65 | 68.84 ± 6.45 |
| World aquatics points | 682.13 ± 58.96 | 729.33 ± 41.5 |
| 50 m backstroke (s) | 27.97 ± 0.81 | 31.3 ± 1.64 |

was set to 800–2,400 m, including technical and speed exercises. After the warm-up, the participants dried their skin and marked their bony landmarks, including the greater trochanter, both acromion processes, and the lateral prominence of the elbow (radius) and wrist (ulna) joints, with a black marker. The study used a within-subject design to complete two 25-m maximum effort backstrokes, first testing the side arm, followed by the top arm breakout technique (Fig. 1). Additional retest opportunities were provided to participants who believed they had insufficient control over their movements during the breakout phase. Ample resting time was provided between each test, with a recommended break of at least 3 min. During this time, we recorded the subject recovery status. To minimize the impact of swimmers' block or turn movements, as per previous studies (*Stosic et al., 2020*; *Trinidad et al., 2020*), the participants were asked to perform a push-off start for the test. The kinematic parameters were recorded from the start (after the push-off), till the head reached the 15-m range.

## Data collection and processing

Before testing, the pool lane lines were removed to ensure unobstructed movement. The study used four underwater cameras (sampling frequency: 60 Hz; resolution: 1,920 × 1,080; GoPro HERO8), of which three were positioned laterally and perpendicular to the swimmers' trajectory at 12.5 m (in the same direction, spaced 5 m apart), mainly to capture segmental two-dimensional (2D) kinematic parameters. Meanwhile, one camera was placed at 17.5 m along the trajectory to capture the 2D angle parameters of the two techniques during critical moments. As the body depth during the backstroke breakout phase was ~0.52 m from the water surface (*Stosic et al., 2020*), with the swimmers generally at a higher position during the session, the camera was positioned 0.8 m above the water surface to capture the 2D angle parameters during the breakout phase. The frame of calibration was located at a position of 12.5 m along the trajectory of the subjects (camera setup for the analysis is shown in Fig. 2). An external synchronization signal light was installed within the range where all four cameras could capture the action to ensure that the movements were measured at exact the same moment.

In the case of a retest, the researchers assessed the subject's performance (based on the 15-m time recorded by camera 3). All the videos were analyzed using the Kinovea (v0.9.5) motion analysis software to obtain the 2D data. While the use of GoPro and Kinovea

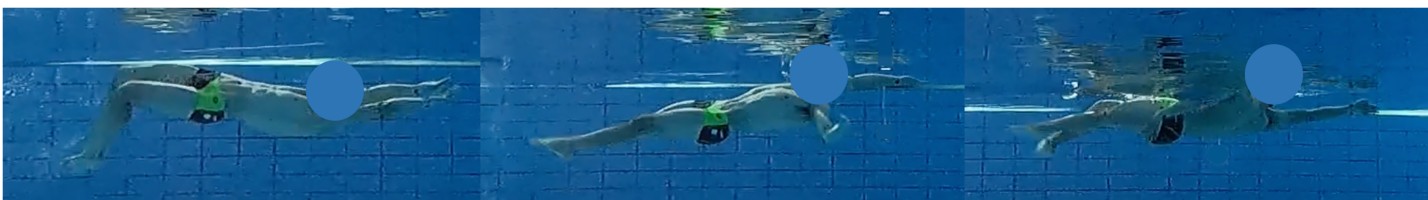

Side Arm

Top Arm

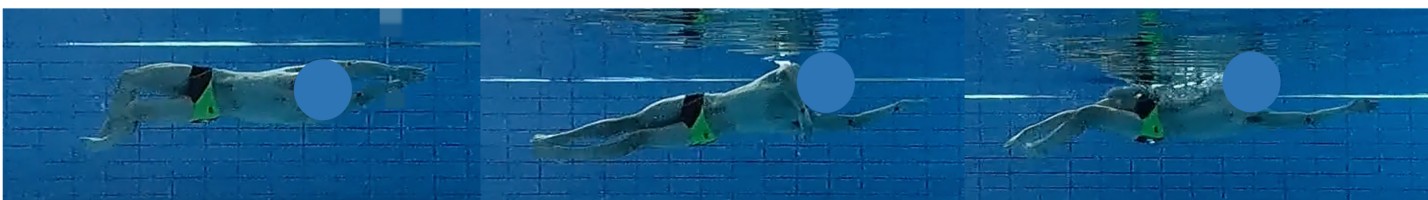

**Figure 1 Schematic comparison of two backstroke breakout techniques.** From left to right are the start, middle and end moments of the two techniques. The three pictures above show the normal side arm technique for backstroke, where the swimmer will stroke from the side of the body. The bottom three pictures show the top arm technique, where the swimmer strokes from the top of the body.

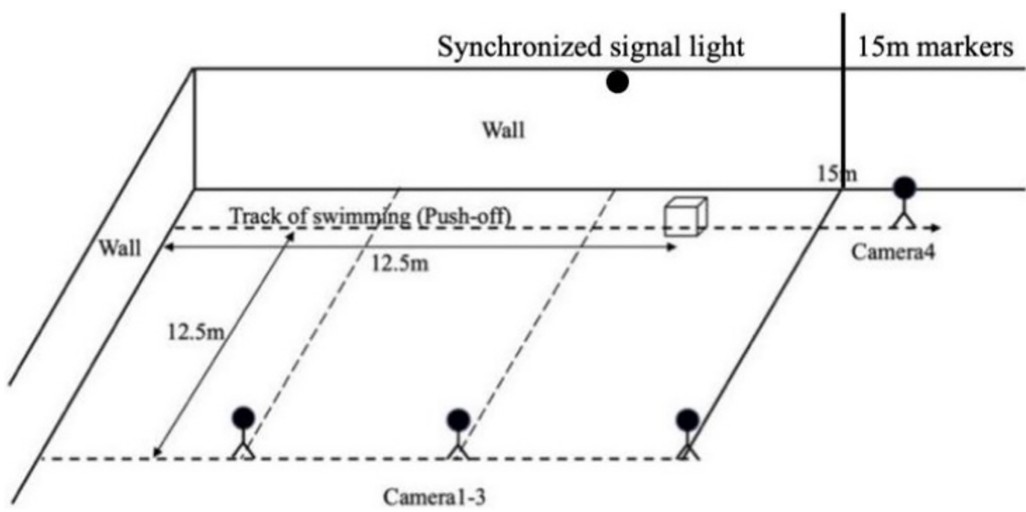

**Figure 2 Schematic diagram of the test site setup and camera positions.**

software to capture and analyze movement data, respectively, may not be optimal due to the study's focus on keyframes and 2D movements rather than continuous frames, the potential impact is relatively small, and the reliability of the equipment and software has been inspected to some extent (*Faelli et al., 2021*; *Edriss et al., 2024*).

## Stages and variables definition

Based on relevant research and our study objectives, the 15-m swim was divided into three sub-stages, namely underwater propulsion (S1), breakout (S2), and clean swimming (S3).

Underwater propulsion included gliding and dolphin kicks, starting from knee extension during the push-off and ending with the last underwater dolphin kick (changing to flutter kick). Breakout (S2) included side arm and top arm breakout techniques, starting from the separation of hands and ending with the emergence of hands from the water. Lastly, clean swimming (S3) began from the end of stage S2 and ended when the swimmer reached the 15-m point. The vertical main optical axis position of camera 3 was located at 15 m; this position was marked on the side of the pool with a vertical pole to determine the swimmer head position. Combined with the push-off time judged by camera 1, a 15-m performance was derived for each swimmer. The time and displacement of each sub-stage were measured using the participant's head as the reference to calculate the average velocity.

Due to the distinct movements of the two breakout techniques, we measured the 2D angle of the hand movement when the hand was located in the vertical plane of the shoulder joint; this was considered a critical moment for dividing the backstroke (*Fernandes et al., 2022*). Indicators used to assess body posture (*Stosic et al., 2020*) included trunk inclination angle (TA): the angle between the shoulder-hip line and the horizontal; body inclination angle (BA): the angle between the shoulder-knee line and the horizontal; shoulder angle (SA): the angle between the two acromion processes and the radius; and elbow angle (EA): the angle between the acromion process and the radius. At the intermediate moments of the breakout phase, the same frames in which cameras 3 and 4 capture the swimmer's actions were judged according to the signaling lamps.

### Statistical analysis

Statistical analysis was performed using SPSS software (v.25, IBM; SPSS Inc., Chicago, IL, USA). Shapiro-Wilk test was used to examine the normal distribution of sample differences, and Levene's test was used for homogeneity of variance. Two-way ANOVA was used to compare the 15-m performance, segmental velocities, and critical moment angles between different genders and breakout techniques. The value of the effector $\eta^2$ is taken as no effect if $0 < \eta^2 < 0.04$; minimum if $0.04 < \eta^2 < 0.25$; moderate if $0.25 < \eta^2 < 0.64$; strong if $\eta^2 > 0.64$ (*Ferguson, 2009*). Pearson correlation analysis was used to explore the relationship between segmental velocities and 15-m performance, with an absolute value of R greater than or equal to 0.6 considered to be a strong correlation. To further analyze the impact of breakout velocity on 15-m performance, hierarchical regression analysis was conducted. The 15-m performance was the dependent variable, with hierarchical regression analysis controlling for breakout velocity as the first independent variable, underwater propulsion phase velocity, and gender as the second independent variables. The significance level $p$ was set at 0.05 and 0.01.

### RESULTS

Table 2 provides data on the mean and standard deviation of speeds at various stages, critical moment angles, and 15-m times for the gender and breakout groups, and Table 3 provides the results of the two-way ANOVA.

For the S2 stage, the male and female swimmers exhibited side arm velocities of $1.51 \pm 0.23$ and $1.29 \pm 0.19$ m/s, respectively, and top arm velocities of $1.72 \pm 0.20$ and

**Table 2 Descriptive statistics of speed and 15-m performance in each stage.**

| Parameters | Gender | Backstroke breakout | |
|---|---|---|---|
| | | Side arm | Top arm |
| V_S1 (m/s) | Male | 1.88 ± 0.09 | 1.88 ± 0.10 |
| | Female | 1.79 ± 0.14 | 1.81 ± 0.12 |
| V_S2 (m/s) | Male | 1.51 ± 0.23 | 1.72 ± 0.20 |
| | Female | 1.29 ± 0.19 | 1.47 ± 0.16 |
| V_S3 (m/s) | Male | 1.74 ± 0.15 | 1.77 ± 0.12 |
| | Female | 1.56 ± 0.13 | 1.51 ± 0.12 |
| T15 (s) | Male | 8.11 ± 0.40 | 7.96 ± 0.25 |
| | Female | 8.71 ± 0.46 | 8.52 ± 0.56 |
| TA (°) | Male | 6.63 ± 5.26 | 5.61 ± 6.02 |
| | Female | 9.51 ± 6.21 | 4.96 ± 5.17 |
| IA (°) | Male | 11.97 ± 9.26 | 69.59 ± 12.71 |
| | Female | 11.73 ± 7.92 | 45.39 ± 21.53 |
| SA (°) | Male | −31.39 ± 8.75 | 110.09 ± 8.19 |
| | Female | −41.31 ± 9.76 | 103.89 ± 14.48 |
| EA (°) | Male | 119.54 ± 7.83 | 105.14 ± 14.87 |
| | Female | 112.86 ± 10.02 | 103.83 ± 17.64 |

**Table 3 Two-way ANOVA results of speed and 15-m performance in each stage.**

| Source of variation | Variable | $F$ | $p$ | $\eta^2$ | Variable | $F$ | $p$ | $\eta^2$ |
|---|---|---|---|---|---|---|---|---|
| Intercept | V_S1 | 7,966.392 | 0.000** | 0.996 | V_S2 | 1,851.289 | 0.000** | 0.985 |
| Sex | | 3.659 | 0.066 | 0.116 | | 11.189 | 0.002** | 0.286 |
| Stroke | | 0.009 | 0.927 | 0.000 | | 8.014 | 0.008** | 0.223 |
| Sex * Stroke | | 0.053 | 0.819 | 0.002 | | 0.037 | 0.849 | 0.001 |
| Intercept | V_S3 | 5,035.976 | 0.000** | 0.994 | T15 | 10,975.956 | 0.000** | 0.997 |
| Sex | | 21.115 | 0.000** | 0.430 | | 13.442 | 0.001** | 0.324 |
| Stroke | | 0.052 | 0.821 | 0.002 | | 1.186 | 0.285 | 0.041 |
| Sex * Stroke | | 0.766 | 0.389 | 0.027 | | 0.016 | 0.900 | 0.001 |
| Intercept | TA | 43.400 | 0.000** | 0.608 | IA | 186.176 | 0.000** | 0.869 |
| Sex | | 0.301 | 0.588 | 0.011 | | 5.780 | 0.023* | 0.171 |
| Stroke | | 1.887 | 0.180 | 0.063 | | 80.641 | 0.000** | 0.742 |
| Sex * Stroke | | 0.763 | 0.390 | 0.027 | | 5.557 | 0.026* | 0.166 |
| Intercept | SA | 333.345 | 0.000** | 0.923 | EA | 2,153.156 | 0.000** | 0.987 |
| Sex | | 4.341 | 0.046* | 0.134 | | 0.707 | 0.408 | 0.025 |
| Stroke | | 1,372.507 | 0.000** | 0.980 | | 6.063 | 0.020* | 0.178 |
| Sex * Stroke | | 0.232 | 0.634 | 0.008 | | 0.320 | 0.576 | 0.011 |

Notes:
* $p < 0.05$.
** $p < 0.01$.

1.47 ± 0.16 m/s, respectively. A significant difference was observed between the (i) breakout techniques ($F = 8.014$, $p = 0.008$, $\eta^2 = 0.223$), indicating a minimal effect; and (ii) genders ($F = 11.189$, $p = 0.002$, $\eta^2 = 0.286$), indicating a moderate effect. However, there was no prominent interaction between genders and breakout techniques ($F = 0.037$, $p = 0.849$, $\eta^2 = 0.001$). At the T15 stage, the male and female swimmers initiated the side arm technique at 8.11 ± 0.40 and 8.71 ± 0.46 s, respectively, and the top arm technique at 7.96 ± 0.25 and 8.52 ± 0.56 s, respectively. The results revealed a substantial difference between genders ($F = 13.442$, $p = 0.001$, $\eta^2 = 0.324$), indicating a moderate effect size, but a negligible difference between the breakout techniques ($F = 1.186$, $p = 0.285$, $\eta^2 = 0.041$). Furthermore, no visible interaction was observed between genders and breakout techniques ($F = 0.016$, $p = 0.900$, $\eta^2 = 0.001$). Moreover, both male and female swimmers demonstrated stable performances in the S1 and S3 stages while using different breakout techniques.

Among the angle parameters, IA, SA, and EA varied extensively between genders and breakout techniques, with IA revealing an interactive effect between the two. These three indicators could effectively describe the differences between the two breakout techniques.

The 15-m performance of the male and female swimmers differed largely. Thus, the influence of gender on the 15-m performance was avoided by analyzing the results of the two groups separately. The correlation analysis heat map is shown in Fig. 3. Correlation analysis showed that for male swimmers, the 15-m performance was (i) significantly negatively correlated with the S1 ($R = -0.73$, $p = 0.003$) and S3 ($R = -0.78$, $p = 0.001$) velocities; and (ii) moderately negatively correlated with the S2 velocity ($R = -0.447$, $p = 0.109$). Meanwhile, correlation analysis showed that for female swimmers, the S1 velocity ($R = -0.875$, $p = 0.000$) and S2 velocity ($R = -0.555$, $p = 0.017$) were significantly negatively correlated with T-15. Moreover, for both male and female swimmers, the S2 phase velocity was moderately/significantly positively correlated with the breakout technique (binary variable).

Hierarchical regression analysis was performed with 15-m performance as the dependent variable, S2 velocity as the first independent variable, and S1 velocity and gender as the second independent variables (Table 4). The model with S2 as the first independent variable exhibited an $R^2$ value of 0.414, indicating that the breakout phase velocity could explain 41.4% of the variation in the 15-m performance. This model passed the $F$-test ($F = 21.216$, $p = 0.000$), suggesting that the breakout phase velocity would have an impact on the 15-m performance. The model formula is: $15T = 10.435 - 1.400 \times S2V$. The model with S1 velocity and gender as independent variables exhibited a significant change in the $F$-value ($p = 0.000$), indicating that the addition of these two independent variables was meaningful to the model. Moreover, the $R^2$ value of the model increased from 0.414 to 0.798, indicating that the combined variable could explain 38.3% of the variation in the 15-m performance. The regression coefficients for S1 velocity and gender were $-3.156$ ($t = -6.476$, $p = 0.000$) and 0.303 ($t = 2.996$, $p = 0.006$), respectively, showing that S1 velocity (gender) had a significant negative (positive) effect on the 15-m performance.
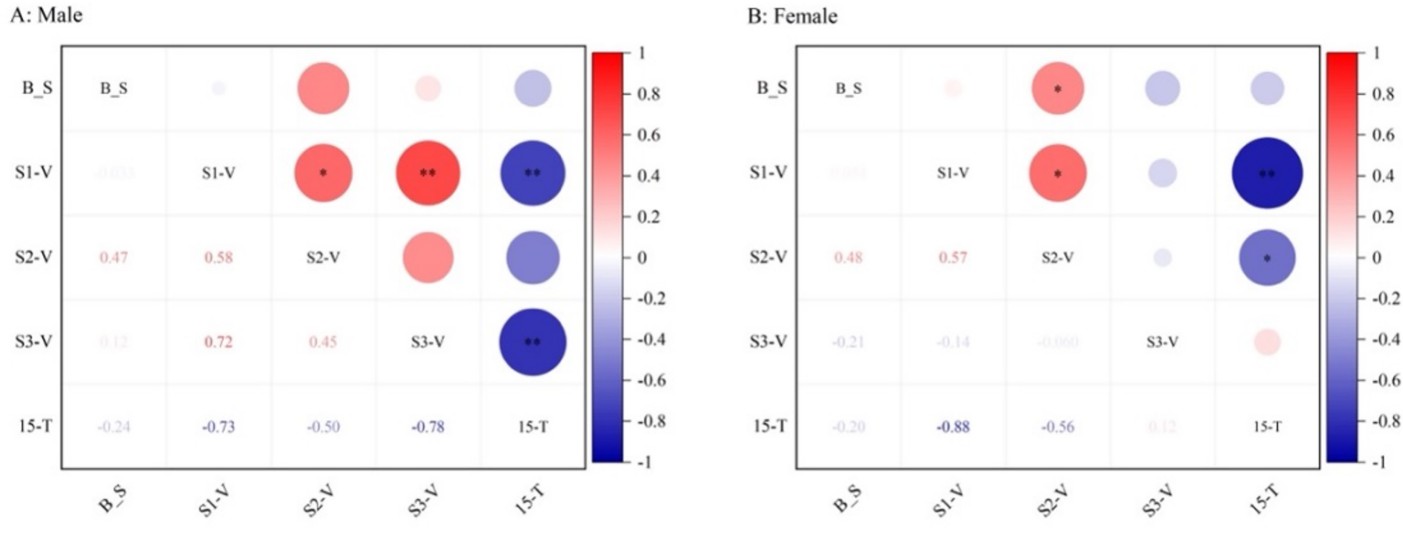

**Figure 3 Analysis of the correlation between speed and 15-m performance of male and female swimmers at each stage.** B_S denotes two Breakout techniques for backstroke, dichotomous variables. S1-V represents the underwater propulsion phase velocity. S2-V represents the breakout phase velocity. S3-V represents the cleaning swimming phase speed. 15-T indicates 15 m performance.

**Table 4 Hierarchical regression results.**

|  | Hierarchy 1 |  |  |  |  | Hierarchy 2 |  |  |  |  |
|---|---|---|---|---|---|---|---|---|---|---|
|  | B | SE | t | p | β | B | SE | t | p | β |
| Constant | 10.435** | 0.456 | 22.903 | 0.000 | – | 13.877** | 0.807 | 17.194 | 0.000 | – |
| S2-V | −1.400** | 0.304 | −4.606 | 0.000 | −0.644 | −0.132 | 0.255 | −0.517 | 0.609 | −0.061 |
| S1-V |  |  |  |  |  | −3.156** | 0.487 | −6.476 | 0.000 | −0.706 |
| Sex |  |  |  |  |  | 0.303** | 0.101 | 2.996 | 0.006 | 0.292 |
| $R^2$ | 0.414 |  |  |  |  | 0.798 |  |  |  |  |
| Adjust $R^2$ | 0.395 |  |  |  |  | 0.776 |  |  |  |  |
| F | F (1,30) = 21.216, p = 0.000 |  |  |  |  | F (3,28) = 36.803, p = 0.000 |  |  |  |  |
| $\Delta R^2$ | 0.414 |  |  |  |  | 0.383 |  |  |  |  |
| $\Delta F$ | F (1,30) = 21.216, p = 0.000 |  |  |  |  | F (2,28) = 26.536, p = 0.000 |  |  |  |  |

**Notes:**
* $p < 0.05$.
** $p < 0.01$.

# DISCUSSION

Our results showed that the breakout speed of the top arm technique was significantly faster than that of the side arm technique irrespective of the gender of the swimmers. The difference in speed between different breakout techniques for the same stroke can reach close to 0.2 m/s, and such a difference may only be seen in different strokes, for example, the difference between backstroke and freestyle is approximately 0.17 or 0.20 m/s (*Stosic et al., 2023*; *Trinidad et al., 2020*).

Several studies have been conducted on the breakout phase of different swim strokes. *Trinidad et al. (2020)* revealed that the backstroke breakout phase had the longest duration and displacement, but the slowest speed (0.20 and 0.15 m/s slower than freestyle and butterfly strokes, respectively). They further observed that the speed of female swimmers during the backstroke breakout phase was 0.10 m/s slower than that during the butterfly-stroke breakout phase (*Trinidad et al., 2022*). *Stosic et al. (2023)* studied national-level male swimmers and found the backstroke breakout speed to be 1.61 m/s, *i.e.*, significantly slower than that of the freestyle (1.78 m/s) and butterfly (1.80 m/s) strokes. *Navandar, Veiga & Navarro (2016)* found that the backstroke breakout speed of international-level junior swimmers was 0.09 m/s slower than their freestyle breakout speed. These reports show that a 0.2 m/s difference in the breakout speed for the same swimmer can be quite significant.

Various studies have demonstrated that the 15-m performance (including start and turn) of the swimmers is strongly correlated with their overall performance in high-level competitions (*Morais et al., 2022b*). Good performance in the individual phases of the race results in a better overall performance of the swimmer, with the initial 15 m of a 50-m event being the most consequential phase (*Morais et al., 2022c*).

The difference in the 15-m performance of the two breakout techniques was ~0.15–0.19 s, indicating that the difference in their breakout speeds affects the 15-m performance of the swimmer. However, the contribution of this effect to the 15-m performance is debatable and likely comes into play only among swimmers of comparable ability. *García-Ramos et al. (2015)* observed differences of ≥1 s in the start phase and ~2 s in the turn phase between athletes of varying levels. Differences in the breakout speed can have a noticeable effect on the 15-m performance, but this effect may not always be decisive, depending on the disparities in the abilities of the swimmers (*García-Ramos et al., 2015*).

Regardless of the employed breakout technique, the speed advantage of the underwater dolphin kick and its impact on the 15-m performance was further emphasized in this study (*Gonjo & Olstad, 2020*). The results showed that among the three segments of the 15-m race, the speed in the underwater propulsion stage was the fastest, while that in the breakout stage was relatively reduced. Moreover, we observed a gradual decline in the overall 15-m performance. In many studies, the breakout phase is usually included in other phases before and after water emergence, however, high-level swimmers typically slow down as they transition from the push-off phase to the clean swimming phase (*Takeda et al., 2009*; *Seifert, Chollet & Chatard, 2007*; *Gonjo & Olstad, 2020*; *Morais et al., 2022a, 2022c*). Therefore, the breakout phase plays a crucial role in preserving the speed advantage gained in the underwater phase, minimizing/maintaining speed, and accomplishing a smooth underwater to above-water transition.

However, certain studies have reported different findings. A study analyzing the swimming world championship found that, except for the women's breaststroke event, the breakout speed (1.61 ± 0.19 m/s) was faster than the clean swimming phase (1.46 ± 0.13 m/s), as well as the underwater phase (1.56 ± 0.19 m/s) (*Veiga & Roig, 2017*). This study, featuring the world's top swimmers, primarily collected data throughout the race, necessitating further analysis of the segmented performance. Some studies have shown that the speed of the

backstroke start phase of female swimmers continues to decline, while the backstroke breakout phase speed of male swimmers ($1.75 \pm 0.13$ m/s) is faster than their leg kicking speed ($1.67 \pm 0.09$ m/s) and surface swimming speed ($1.17 \pm 0.34$ m/s) and slower than gliding speed ($2.21 \pm 0.32$ m/s) (*Trinidad et al., 2022*). Relevant reports show inconsistent findings, possibly varying due to factors, such as the skill level of the subjects, type of start (*e.g.*, push-off), data collection methods, and stage division methods (*Gonjo & Olstad, 2021*). Our study found no significant differences in the speed of the underwater propulsion stage of the two techniques, consistent with the previous research.

Elite swimmers demonstrate significantly better techniques, exhibiting a more symmetrical backstroke body roll compared to others(*Barbosa et al., 2010*; *Cappaert, 1999*). Alternating stroke is a basic characteristic of the arm movement pattern of backstroke and freestyle; high-level swimmers generally make full use of body rotation and control during the stroke to improve swimming speed and efficiency (*Gonjo et al., 2021*). The breakout to some extent is also the underwater part of the first action of the stroke swimming. The difference in the SA can explain the body rotation problem to a certain extent. Generally, the range of motion of the shoulder joint in backstroke and freestyle is greater than that of the hip joint (*Gonjo et al., 2021*; *Riewald & Rodeo, 2015*). The difference in SL and SR can also affect the speed and efficiency of the breakout stage. During backstroke, elite swimmers adjust the timing and range of body rotation and SL and SR to improve stroke efficiency (*Gonjo et al., 2021*). The breakout frequency and SL can also differ for clean swimming techniques. For instance, a difference in SL can lead to changes in the backstroke and freestyle stroke speeds (*Gonjo et al., 2020*). For example, *Stosic et al. (2023)* found that the SL in the breakout phase of backstroke decreased from $2.22 \pm 0.43$ (m/cyc) to $2.08 \pm 0.24$ (m/cyc) in the clean swimming phase, and the SL slowed from $44.72 \pm 7.92$ (m/cyc) to $42.66 \pm 4.82$ (m/cyc). slowed down to $42.66 \pm 4.82$ (m/cyc), consistent with other reports on the competition parameters of the world's top swimmers (*Shen et al., 2015*).

Most swimmers are known to coordinate their arm movements with the last dolphin kick during the breakout in backstrokes and freestyle strokes (*Trinidad et al., 2020*), consistent with our findings. Only two subjects in this study coordinated their side arm technique with the flutter kick, suggesting that the swimmers switched the last dolphin kick to the flutter kick before breakout. As we did not heavily intervene with the training of the subjects, this coordination was a direct result of the speed and technical rhythm of the swimmers.

Apart from the influence of arm technique, body posture, and hand-leg coordination, the swimming trajectory and depth also affected the breakout speed (*Naemi & Sanders, 2008*). Additionally, breakout distance is an important factor in the starting phase, which is known to vary among athletes of different levels in high-level backstroke races (*Morais et al., 2022b*). The impact of the breakout phase is reflected in the subsequent swimming process, and elite swimmers generally demonstrate high control of breakout distance (*Veiga et al., 2014*), depth, and smooth transition.

Although this study provides a reference for the techniques used in the breakout phase of backstroke starts and turns, it also faces certain limitations. Firstly, we were unable to

explicitly quantify the proficiency of the side arm and top arm techniques in this analysis due to the different experience and skill levels of the swimmers. Despite our results showing that the top arm technique is faster than the side arm technique, currently, we are unable to provide a quantitative explanation for the proficiency of the two techniques. Additionally, the data collection and processing method used in this study may not be the most accurate. Moreover, the parameters such as the angle and angular velocity are not very rich and the 2D data exhibit certain limitations. In the future, attempts could be made to collect and analyze related movements using three-dimensional methods. Coaches and athletes need to ensure fluidity of movement during the breakout phase. According to the rules, if an athlete chooses to breakout using the top arm technique, care should be taken not to tilt the body sideways more than 90°.

## CONCLUSIONS

The backstroke top arm technique may have a speed advantage over the traditional side arm technique during the breakout phase, significantly affecting the 15-m performance of the swimmers. However, this advantage may need to be considered in conjunction with the smoothness of the transition to the preceding and subsequent phases. Coaches and swimmers are advised to opt for techniques that correspond to their specific traits and habitual tendencies, applying rational control in their execution.

## ACKNOWLEDGEMENTS

We gratefully acknowledge all the participants for their involvement in the study, as well as the researchers who provided assistance with the experiments and the preparation of the manuscript.

### Funding
This work was supported by the Fundamental Research Funds for the Central Universities (No. 2024YDXL003). The funders had no role in study design, data collection and analysis, decision to publish, or preparation of the manuscript.

### Grant Disclosures
The following grant information was disclosed by the authors:
Fundamental Research Funds for the Central Universities: 2024YDXL003.

### Competing Interests
The authors declare that they have no competing interests.

### Author Contributions
• Zhenyu Jin conceived and designed the experiments, performed the experiments, analyzed the data, prepared figures and/or tables, authored or reviewed drafts of the article, and approved the final draft.

- Yuhang Zhou conceived and designed the experiments, analyzed the data, prepared figures and/or tables, authored or reviewed drafts of the article, and approved the final draft.
- Dapeng Wang conceived and designed the experiments, performed the experiments, authored or reviewed drafts of the article, and approved the final draft.
- Yuhong Wen conceived and designed the experiments, authored or reviewed drafts of the article, and approved the final draft.

## Human Ethics

The following information was supplied relating to ethical approvals (*i.e.*, approving body and any reference numbers):

Approved by The Sports Science Experiments of the Beijing Sport University (Approval number: BSU2023223H).

## Data Availability

The raw measurements are available in the Supplemental File.

## Supplemental Information

Supplemental information for this article can be found online at http://dx.doi.org/10.7717/peerj.18838#supplemental-information.

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
