# Peer review of "The impact of side and top arm techniques during the backstroke breakout phase on 15-meter swimming performance"

_PeerJ, doi:10.7717/peerj.18838_

## Round 0.1 · original submission · Major Revisions

Reviewers consider this as an interesting manuscript but are requesting further clarifications and elaboration on the results interpretation.

·

Basic reporting

the manuscript nees to be proofread

Experimental design

no comment

Validity of the findings

no comment

Additional comments

no comment

·

Basic reporting

The article is well constructed, and it give us some insides of the techniques of backstroke breakout. This topic interests’ researchers but also coaches in order to optimize the individual performance of swimmers

Authors can add some references to improve the discussion part

General Remarks
Please better define the two techniques side arms and top arms at the process and add that in the legend of the figure 1

Abstract
Methods
Add the stroke and the event or the word ranking and year of the points of the population, and separate male and female to specify the level.

Experimental design

I had some remarks concerning the methodology and the chosen markers to study the impact of the two swimming breakouts in backstroke.

Materials
(2) the main stroke of the best results of the world aquatics points at least 600 points, please specify in which event ? 50m backstroke ? or in the year ranking
(3) the main or second main event being backstroke or medley in which distance ?
L 102. Please put the number of training by week or the approximate hours if possible.
L 104. The best performance was realized in 50m or 25m pool ? For the swimmers that they did ‘not had a time in 50m backstroke, The test was conducted by researched with other participants or only one swimmer ?
L114. The rest between the two 25m was 3 minutes for all the swimmer ?
How do control if the swimmer control their movements properly during the breakout phase ?
How many swimmer made an additional opportunities of retest ? After how did you chose the test to keep in the study?

Data collection
L. 121: the pool lines were removed to ensure that movements were not obstructed how you measure the 15m time?? I think that you put other marks out of the pool to see when the head of the swimmer pass the 15m. Swimmer continue to swam until 20m? or they stop at 15m?

Researchers verify the position of the feet’s or the position of the body before the push off?

L 135-14. You synchronize the cameras with the light and then you said that this study did not analyze continuous frames that means that research verify the synchronization of the cameras and then you analyze camera by camera?

L 285. The authors remarks that only two subjects coordinated with the flutter kick during the side arm that means that for the top arm all swimmers realized a dolphin kick at the first movement at the breakout at the first arm pull? There is the no body roll through the water surface?

Validity of the findings

Discussion:
Please discuss more about the limitations of the study
In my opinion swimmers will use an optimal and individual strategy to overcome the drag resistance. I think relative underwater distance, start position feet’s (high), deep position for dolphin kicks. These parameters could be interesting to study the individual inter variability of underwater and breakout strategy.
Maybe it could be interesting to indicate the total time of each phase and the number of dolphin kick for each condition.
Barkwell et al 2020 show that Both head entry distance and takeoff velocity are related to start performance, suggesting each position may optimize different aspects of the backstroke start.
In breaststroke there is a study of Gonjo T, Olstad BH, Stastny´ J, Conceicão A, Seifert L (2023) Intra- and inter-individual variability in the underwater pull-out technique in 200 m breaststroke turns. PLoS ONE 18(3): e0283234
They found that in breaststroke both inter- and intra-individual variabilities during the underwater phase were evident in 200 m breaststroke turns, which were categorized into three patterns based on the timing of the dolphin kick and the duration of glides. In some cases intra-individual variability were observed in the relative glide (with the arms at the side) duration and distance.
Authors can also include the study of Chainok et all in the discussion
Chainok et al. 2023 found that backstroke-to-breaststroke turning techniques are specific; developing approaching speed in conjunction with proper gliding posture and pull-out strategy will result in improved turning performance and may influence differently the development of specific training intervention programs.

You can also include practical part to coaches for working the breakout individual adaptation strategies.

·

Basic reporting

The manuscript entitled "The Impact of Side and Top Arm Techniques During the Backstroke Breakout Phase on 15-Meter Performance" aimed to study the effect of Side and Top Arm Techniques During the Backstroke Breakout Phase on 15-Meter Performance. The main data suggest that the top arm technique in backstroke may offer a velocity advantage over the traditional side arm technique during the breakout phase, influencing 15-meter performance. It is a very interesting subject and it fits in the scope of the selected journal, which focuses on essential biomechanics measures to improve backstroke performance in high-level swimmers. However, it must be emphasized that this manuscript requires some corrections before being accepted for publication:

Experimental design

This section is well structured and clear

Validity of the findings

All results are valid and clear

Additional comments

Abstract
The abstract is well structured and presents the important information of the article.
L22. "Swimmers" Please change the letter S to lowercase form
Introduction
I noticed that the introduction is composed of 3 paragraphs and that paragraph 2 is long (15 lines). Please develop the introduction (4 to 6 paragraphs).
L61-65. Please provide more information regarding how the importance of the breakout phase can enhancing performance in the start and turn phases and even the overall race time for high-level swimmers (provide statistical values provided in previous studies).
L66-68. “This significance is not only reflected in breakout velocity but can also impact the rhythm of subsequent swimming movements (Veiga S & Roig A et al., 2017; Shen Y et al., 2015) and even psychological state (Wen et al., 2005).” Please expand on these ideas by presenting statistical values.
Materials, Methods, and Results
The materials and methods and results sections are well structured and clear.
Discussion
L223-242. Please present your findings from your statistical analysis and discuss them with the results of previous studies.
L227-228. Please avoid redundancy, you already wrote this paragraph in the introduction section.
L243-267. I noticed that this paragraph is very long (25 lines), I recommend that you rewrite it and present only what is necessary.
L243-249. How do these findings from previous studies relate to your results?

---

## Round 0.2 · accepted · Accept

Reviewers are happy with this new version of the manuscript.

I would like to bring authors attention that in-text citations and references styles must be edited before publication.

PeerJ uses the 'Name. Year' style with an alphabetized reference list.
In-text citations
For three or fewer authors, list all author names (e.g. Smith, Jones & Johnson, 2004). For four or more, abbreviate with ‘first author’ et al. (e.g. Smith et al., 2005).
Multiple references to the same item should be separated with a semicolon (;) and ordered chronologically.
References by the same author in the same year should be differentiated by letters (Smith, 2001a; Smith, 2001b).

The Reference Section

Journal reference format: List of authors (with initials). Publication year. Full article title. Full title of the Journal, volume: page extents. DOI (if available).

·

Basic reporting

The authors have improved the manuscript based on the comments addressed, which is now clearer for readers.

Experimental design

improved and well defined

Validity of the findings

improved and better discussed

·

Basic reporting

Congratulations to the authors for taking into account some remarks as well as the explanations on others.
The added litterature help to improuve the discussion and findings

Experimental design

Methods was now detailed

Validity of the findings

The findings help to understand the two techniques of backstroke breakout and give some insight to practitioners or coachs

·

Basic reporting

The author has responded rigorously and in detail to the initial critiques, and the revised manuscript is now complete and ready for acceptance.

Experimental design

The author has responded rigorously and in detail to the initial critiques.

Validity of the findings

The author has responded rigorously and in detail to the initial critiques.

Additional comments

The author has responded rigorously and in detail to the initial critiques.